# Golden Apples or Green Apples? The Effect of Entrepreneurial Creativity on Green Entrepreneurship: A Dual Pathway Model

**Hui Jiang [1,2], Suli Wang [2], Lu Wang [2] and Gang Li [3,\*]**

[1] Department of Marketing, School of Management, Zhejiang University, Hangzhou 310058, China; jianghui@zufe.edu.cn

[2] Department of E-commerce, School of Information Management and Artificial Intelligence, Zhejiang University of Finance and Economics, Hangzhou 310018, China; wangsuli97@zufe.edu.cn (S.W.); wanglu@zufe.edu.cn (L.W.)

[3] Shandong Computer Science Center (National Supercomputer Center in Jinan), Qilu University of Technology (Shandong Academy of Sciences), Jinan 250000, China

\* Correspondence: lig@sdas.org

**Abstract:** Entrepreneurs with high creativity (i.e., golden apples) are easy to find, but entrepreneurs with green entrepreneurial intention (i.e., green apples) are rare. To explain this phenomenon, we first introduce cognitive dissonance theory to demonstrate how entrepreneurial creativity influences green entrepreneurship through two parallel mechanisms—green recognition and green disengagement. Moreover, we propose the use of green self-identity as a moderator to predict when the relationships between entrepreneurial creativity and these two mechanisms are intensified or attenuated. Through an empirical study, we surveyed 362 entrepreneurs from a local entrepreneurship association in eastern China. The results show that entrepreneurial creativity is positively associated with both green recognition and green disengagement. While green recognition strengthens green entrepreneurial intention, green disengagement weakens green entrepreneurial intention. More importantly, creative entrepreneurs with high green self-identity are more likely to engage in green recognition and, thus, promote green entrepreneurial intention. By contrast, creative entrepreneurs with low green self-identity are more willing to engage in green disengagement and, thus, inhibit green entrepreneurial intention. Finally, we discuss the theoretical and practical implications of these findings for entrepreneurial creativity and green entrepreneurship.

**Keywords:** entrepreneurial creativity; green recognition; green disengagement; green self-identity; green entrepreneurship

---

## 1. Introduction

The current research investigates why and how creative entrepreneurs engage in opposite decisions for green entrepreneurship (also called sustainable entrepreneurship) [1]. We maintain that creativity should not be solely treated as a driving force for green entrepreneurial behaviors, but that the impact of creativity is malleable, and can vary from positive to negative depending on the personal attitudes toward, and concepts of, environmental protection. Therefore, we explore the underlying promoting (e.g., green recognition) and inhibiting (e.g., green disengagement) mechanisms between entrepreneurial creativity and green entrepreneurial intention, as well as the potential moderators (e.g., green self-identity).

Green entrepreneurship is acknowledged as one of the most valuable mechanisms for promoting a green economy [2]. However, green entrepreneurship faces many barriers in the domains of markets,

financing, and ethics [3]. One potential antecedent of green entrepreneurship is creativity, which means that an entrepreneur can think flexibly and divergently, and generate creative ideas for exploring and recognizing new business opportunities [4,5]. Creativity is essential for benefit recognition, which positively influences general entrepreneurial behaviors [6]. Thus, it is plausible that all creative entrepreneurs should naturally develop green entrepreneurship. However, there may be issues if highly creative entrepreneurs make excuses for refusing green entrepreneurship.

The present study proposes a dual pathway model for the relationship between creativity and green entrepreneurial intention—a unified framework describing the reasons for green transgressions and green behaviors. Due to their flexible cognition and divergent thinking [4,7], entrepreneurs with high creativity may develop ideas for identifying the benefits of environmental businesses. While the pathway of green recognition, a process for identifying green benefits, explains why creativity can increase the possibility of conducting green entrepreneurial behaviors, the path of green disengagement, a process that avoids green responsibility, describes the opposite effect of creativity. That is, entrepreneurs with high creativity may also use their flexible cognition and divergent thinking to delegate environmental responsibility by removing their own responsibility, adopting double standards and using noble excuses to claim that their actions are justified.

Moreover, we suggest that green self-identity, an individual's self-regulation as a pro-environmental person [8], is a critical moderator in determining when creativity is associated with the two pathways above. As the environmental self-regulation effect only works when an individual pays attention to environment-relevant information [9], green self-identity is a critical indicator of when green recognition or green disengagement could be activated by an entrepreneurial creativity. In other words, we propose that activation of the mechanisms of green recognition or green disengagement relies on the level of green self-identity. The underlying reason is that green self-identity enables entrepreneurs to treat "green" as inseparable from their own self-beliefs; thus, they will act in ways that agree with their green self-concept. Hence, creative entrepreneurs who are high in green self-identity will tend to utilize their flexible cognition and divergent thinking to identify potential benefits from new pro-environmental business ventures. On the other hand, creative entrepreneurs who have low green self-identity will tend to utilize their flexible cognition and divergent thinking to delegate their sense of responsibility to protect the environment. Figure 1 shows our dual pathway model, which depicts how and when entrepreneurial creativity affects their green entrepreneurial intention.

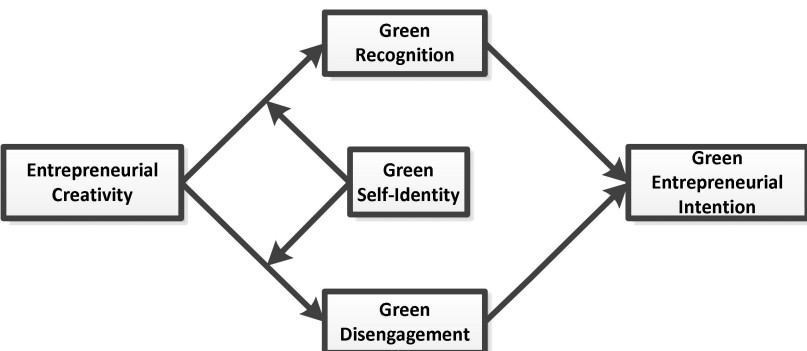

**Figure 1.** The dual pathway model of green entrepreneurial intention.

The present study makes several contributions. First, it explains why not all entrepreneurs with high creativity are willing to start a pro-environmental business venture. We use cognitive dissonance theory [10] to explain the dual pathway model for green entrepreneurial behaviors by simultaneously examining the inhibiting mechanism of green disengagement and the promoting mechanism of green recognition. Second, an essential moderator, the green self-identity, is investigated in terms of when the inhibiting mechanism of green disengagement operates and when the promoting mechanism of green recognition works. The results show that individual characteristics are a potential factor for

determining green entrepreneurial intention. Third, this paper extends the literature of creativity into the field of green entrepreneurship in a more nuanced way. While existing research has widely focused on the positive relationship between creativity and entrepreneurial behaviors [11,12], our results provide evidence that creativity can have a negative impact on green entrepreneurship.

## 2. Theoretical Background and Development of Hypotheses

### 2.1. Entrepreneurial Creativity

As entrepreneurs have to be creative enough to explore and recognize opportunities for new ventures, creativity is widely acknowledged as a critical determinant of entrepreneurial behaviors [13,14]. Many studies in the domain of entrepreneurship have explored creativity but, so far, no uniform definition has emerged. For example, Amabile [15] defines creativity as the production of novel and useful ideas. Based on this definition, Ip, Liang, Wu, Law and Liu [12] further assessed creativity from the two dimensions of originality and usefulness. However, other researchers have regarded creativity as an internal resource to generate innovative methods [16,17]. Specifically, it is relatively common for creative individuals to engage in flexible cognition and divergent thinking processes, which act as critical internal resources in the whole creative process [18].

As a particular group, entrepreneurs have similar cognitive attributes and internal resources, such as flexible cognition and divergent thinking, which make them different from others [4]. Thus, the present study defines entrepreneurial creativity as the internal resource that entrepreneurs use to engage in flexible cognition and divergent thinking. However, these creative attributes only make entrepreneurs more likely to use flexible recognition and divergent thinking to resolve their problems, whether positive or negative. For example, when faced with a difficulty, some people will use their creative thinking to resolve this, while others will attempt to make excuses to avoid the issue. Hence, we propose that there are two opposing mechanisms determining the effect of entrepreneurial creativity on their green entrepreneurial intention.

### 2.2. Green Entrepreneurship and Cognitive Dissonance Theory

Green entrepreneurship refers to a set of entrepreneurial activities that create economic and environmental values by delivering green products and services [19–22]. Compared to traditional commercial entrepreneurship, green entrepreneurship focuses not only on economic returns but, also, the natural environment and sustainable development [23]. Therefore, green entrepreneurs face a multitude of barriers. Linnanen [3] proposed a framework for the boundaries of green entrepreneurship, sorting the edges into three main categories: market barriers (i.e., the lack of consumer awareness of environmental protection), financing barriers (i.e., a high investment cost and lack of funding), and ethical barriers (i.e., a lack of attitude and moral reasoning toward environment protection). Hence, entrepreneurs face a dilemma when considering whether to start a green business venture. Starting such a venture requires them to take high risks because of the abovementioned barriers. In choosing otherwise, they will suffer from self-sanctions due to the lack of green consciousness.

Cognitive dissonance theory provides a concise framework to predict how entrepreneurial creativity can be associated with green entrepreneurship [24]. According to cognitive dissonance theory, when individuals have to express their views on an emerging factor, they experience a psychological conflict between the new cognition and the old cognition, which is called cognitive dissonance [10]. For example, if an entrepreneur holds the two cognitions that "Green entrepreneurship contributes to environmental protection" and "Green entrepreneurship is highly risky", he or she would feel cognitive dissonance. To eliminate the discomfort associated with this conflict, individuals tend to adopt one of two pathways of self-regulation to obtain psychological balance. One is to deny the new cognition directly, and the other is to seek information about the new cognition to replace the old cognition completely. Continuing the example above, the entrepreneur may deny the former cognition: "Green entrepreneurship may not be unnecessary to benefit the environment". On the other

hand, the entrepreneur may also deny the latter cognition: "Although green entrepreneurship is high risk, it is also very profitable".

### 2.3. The Mechanism of Green Recognition

Based on cognitive dissonance theory, we predict that when faced with green entrepreneurship as an inescapable problem, entrepreneurs would feel a psychological conflict between self-sanctions and the high risks arising from the green dilemma. These entrepreneurs can then behave (i) positively in green entrepreneurship by promoting environment protection or (ii) negatively in green entrepreneurship by escaping environmental responsibility.

To examine our above predictions, we initially introduce green recognition as a mechanism for interpreting why entrepreneurs with high creativity may be motivated to start a green business venture positively. Green recognition refers to a form of specific reasoning that represents an individual's ability to identify and discover potential benefits of green entrepreneurship [25]. The existing literature emphasizes the need to resolve discomfort for an individual with cognitive dissonance. One effective way to resolve this dissonance is to actively find abundant evidence to support an action [26,27]. Researchers highly appreciate the role of creativity in the domain of global opportunity recognition [12]. As creative entrepreneurs possess flexible cognition and divergent thinking [18], they can generate many ideas to support their green behaviors.

Furthermore, green recognition should be positively associated with green entrepreneurial intention because it can motivate entrepreneurs to strengthen the green benefits and weaken the green risks by (a) identifying and discovering the potential benefits of a green business venture, (b) being aware of the environmental implications of potential green decisions, (c) evaluating old psychological conflicts (whether they exist here or elsewhere), and (d) creatively reframing cognitions, and generating novel and useful solutions to green dilemmas.

Thus, green recognition enables entrepreneurs to resolve the green dilemma by strengthening their cognition of the benefits of green entrepreneurship. Furthermore, green recognition can also improve entrepreneurs' environmental awareness and green self-concept and, thus, enhance their green entrepreneurial intention [28].

**H1:** *Entrepreneurial creativity is positively associated with green recognition.*

**H2:** *Green recognition is positively associated with green entrepreneurial intention.*

We do not maintain that creativity directly contributes to green entrepreneurial intention, as creativity itself does not involve green issues. Although creativity promotes identification of benefits for both traditional commercial and green businesses [29], the creative process of entrepreneurship does not primarily concern green issues, and creativity does not necessarily result in green behaviors. Moreover, the existing literature shows that environmental self-regulation is not activated by environmental commitment but by green motivation [9,30]. Therefore, we introduce the notion of green self-identity as a moderator. Green self-identity is a cognitive representation of the green self-identity that represents the extent to which green traits are essential to one's self-concept [31,32]. This identity affects individuals' green behavior and remains relatively steady over time [33]. For example, individuals who view themselves as typical garbage sorters are more likely to implement garbage classification than those who do not [34].

Green recognition leads entrepreneurs to focus on environmental issues, identify potential green benefits, and discover the meaningfulness and impact of potential green entrepreneurial behaviors on the environment. Self-concept maintenance theory suggests that individuals prefer to behave consistently with their salient identities [35,36]. It is not a stretch to infer that entrepreneurs with high green self-identity will be more strongly motivated and likely to make green-related decisions than those with low green self-identity. Hence, we predict that entrepreneurs with high creativity and high green self-identity are more likely to engage in green recognition. Specifically, when faced with green dilemmas, entrepreneurs can use their flexible cognition and divergent thinking to fully

engage in green recognition to identify and discover the potential benefits of green entrepreneurship. However, when creative entrepreneurs have low green self-identity, they have less motivation to use their flexible cognition and divergent thinking to engage in green recognition. In other words, creative entrepreneurs may have the ability to resolve the green dilemmas, but they may also lack strong motivation to activate their mechanisms of green recognition.

**H3:** *Green self-identity will moderate the relationship between entrepreneurial creativity and green recognition. Specifically, when green self-identity is high (low), higher entrepreneurial creativity will lead to higher (lower) green recognition.*

Furthermore, we suggest that green recognition mediates the interaction effect between entrepreneurial creativity and green self-identity on green entrepreneurial intention. As mentioned, creativity is positively (negatively) related to the green recognition process when interacting with high (low) green self-identity. Meanwhile, the activation of green recognition mechanisms will lead to higher green entrepreneurial intention. Likewise, green entrepreneurial behavior is a result of personal characteristics (creativity), internal standards (green self-identity), and the process of self-regulation (green recognition). Hence, we predict that when green self-identity is high (low), the indirect effect of creativity on green entrepreneurial intention via green recognition will be more (less) positive.

**H4:** *Green self-identity moderates the indirect effect of entrepreneurial creativity on green entrepreneurial intention via green recognition.*

### 2.4. The Mechanism of Green Disengagement

In contrast to green recognition, we introduce green disengagement as an alternative mechanism to interpret why entrepreneurs with high creativity may be motivated to decline a green business venture. Prior literature has extensively studied the individual behavior of disengagement in various fields, such as moral disengagement [37,38], cultural disengagement [39], and civic disengagement [40]. We first introduce the notion of disengagement into the domain of green entrepreneurship and define green disengagement as a process of specific recognition by which individuals disconnect from environmental awareness and responsibility. Unlike the mechanism of green recognition, another effective way to resolve the discomfort of an individual with cognitive dissonance is to actively find abundant evidence to justify canceling the action [26,27]. The existing literature has highlighted the dark side of creativity, especially in unethical behaviors [41]. As creative entrepreneurs possess flexible cognition and divergent thinking [18], they can generate multiple ways to justify their non-green behaviors.

Furthermore, green disengagement is negatively associated with green entrepreneurial intention because it can motivate entrepreneurs to avoid self-sanctions due to a lack of environmental awareness and responsibility by (a) redefining and distorting the meaning of green behaviors through self-justification, (b) minimizing the value of green behaviors, and (c) blurring and extending the scope of green behaviors. The critical role of green disengagement manifests as individuals reconstructing their perceptions of non-green behaviors by ignoring and minimizing their negative effects. That is, green disengagement enables entrepreneurs to resolve the green dilemma by blocking self-regulatory processes related to environmental protection. Furthermore, green disengagement can also weaken entrepreneurs' environmental awareness and green concept, thus inhibiting their green entrepreneurial intention.

**H5:** *Entrepreneurial creativity is positively associated with green disengagement.*

**H6:** *Green disengagement is negatively associated with green entrepreneurial intention.*

Consistent with the mechanism of green recognition, we propose that green self-identity is a moderator influencing the relationship between entrepreneurial creativity and green disengagement. Researchers have demonstrated that creative individuals can use their flexible and divergent thinking to develop self-justifications [42]. However, the extent of self-justification is determined by personal

characteristics, such as one's self-concept [43]. To maintain their salient self-concepts (i.e., green role identities), such individuals will be more inclined to engage in self-justification. Hence, we predict that entrepreneurs with high creativity and low green self-identity are more likely to engage in green disengagement. Specifically, when faced with green dilemmas, entrepreneurs can use their flexible cognition and divergent thinking to fully engage in green disengagement to avoid self-sanctions due to a lack of environmental awareness and responsibility. However, when creative entrepreneurs have high green self-identity, they will have less motivation to use their flexible cognition and divergent thinking to engage in green disengagement. In other words, creative entrepreneurs that have the ability to resolve green dilemmas may also lack strong motivation to activate the mechanism of green disengagement.

**H7:** *Green self-identity will moderate the relationship between entrepreneurial creativity and green disengagement. Specifically, when green self-identity is low (high), higher entrepreneurial creativity will lead to higher (lower) green disengagement.*

Furthermore, we suggest that green disengagement mediates the interaction effect of creativity and green self-identity on green entrepreneurial intention. As mentioned, creativity is negatively (positively) related to the green disengagement process when interacting with high (low) green self-identity. Meanwhile, the activation of green disengagement mechanisms will lead to lower green entrepreneurial intentions. Along this line, green entrepreneurial behavior is a result of personal characteristics (creativity), internal standards (green self-identity), and the process of self-regulation (green disengagement). Hence, we predict that when green self-identity is high (low), the indirect effect of creativity on green entrepreneurial intention via green disengagement will be more (less) negative.

**H8:** *Green self-identity moderates the indirect effect of entrepreneurial creativity on green entrepreneurial intention via green disengagement.*

## 3. Methods

### 3.1. Procedures and Participants

We collected data using an online questionnaire distributed to entrepreneurs from different domains. The information of these entrepreneurs came from a database provided by a local entrepreneurship association in eastern China. Before filling out the questionnaire, the participants were informed that their responses were strictly confidential and anonymous. The questionnaire consisted of two parts: demographics (e.g., age, gender, education, work field, and tenure) and measured variables (e.g., creativity, green self-identity, green disengagement, green recognition, green entrepreneurial intention, self-efficacy, and social capital). Participants were randomly selected from the database and invited to participate via an email containing a brief description and a link to our research.

The investigation lasted for four months. In total, 832 invitations were issued, and 362 valid questionnaires were recovered (response rate = 43.51%). The participants consisted of 56.08% males (SD = 0.50) with a mean age of 40.81 years (SD = 7.66). The majority of participants (74.85%, SD = 0.43) hold a bachelor's degree or above and have spent an average of 6.00 years (SD = 4.87) in their current job. The participants came from the following domains: manufacturing (22.65%), information technology (18.23%), mass media (8.84%), agriculture (3.04%), healthcare (5.52%), construction (4.70%), transport (5.52%), accountancy (1.10%), finance (3.31%), tourism (4.42%), retail (8.01%), consulting (2.76%), and other (11.88%).

### 3.2. Control Variables

As self-efficacy is a belief to be able to accomplish specific tasks successfully, entrepreneurs high in self-efficacy are more likely to start up a new venture than entrepreneurs low in self-efficacy [44].

Moreover, the existing literature also proposes that the more social capital (e.g., mentors, informal industry networks, and professional forums) entrepreneurs have, the more likely they are to start up a new venture [44]. Since there are strong correlations between these variables and entrepreneurial practices, we controlled for self-efficacy and social capital to test whether entrepreneurial creativity alone accounted for green entrepreneurship.

### 3.3. Measures

According to the existing literature, all variables were measured using 7-point Likert scales, which were determined to be mature and reliable. To ensure equivalency between languages, we initially designed the scales in English, translated them into Chinese, and then back-translated them into English using professional translators. We measured the entrepreneurial creativity as an independent variable using a 12-item scale ($\alpha$ = 0.89) adapted from Chia and Liang [44] (see Table A1 in Appendix A). Green self-identity was measured as a moderator using a 6-item scale ($\alpha$ = 0.85) adapted from Chen [45] (see Table A2 in Appendix A). Two mediators, green recognition and green disengagement, were measured by a 6-item ($\alpha$ = 0.87) scale adapted from Ozgen and Baron [46] (see Table A3 in Appendix A) and an 8-item scale ($\alpha$ = 0.85) adapted from Moore et al. [47] (see Table A4 in Appendix A). We also measured green entrepreneurial intention as a dependent variable using a 6-item scale ($\alpha$ = 0.84) adapted from Wang et al. [48] (see Table A5 in Appendix A). Self-efficacy was measured by an 8-item scale ($\alpha$ = 0.83) adapted from Wang, Chang, Yao and Liang [48] (see Table A6 in Appendix A), and social capital was measured using a 10-item scale ($\alpha$ = 0.83) adapted from Williams [49] (see Table A7 in Appendix A).

## 4. Analyses and Results

### 4.1. Descriptive Statistics

As there were several green-related focal constructs (i.e., green self-identity, green disengagement, green recognition, and green entrepreneurial intention) in our research, we conducted a confirmatory factor analysis to examine the discriminant validity. The results show that the four-factor model was significantly superior to the other models (Table 1).

**Table 1.** Confirmatory factor analysis for discriminant validity.

| No. | Model | $\chi^2$ | df | $\chi^2/df$ | NFI | CFI | RMSEA | $\Delta\chi^2$ |
|-----|-------|----------|-----|-------------|------|------|--------|----------------|
| 1 | Four-factor model | 334.37 | 293 | 1.14 | 0.91 | 0.99 | 0.02 | |
| 2 | Three-factor model 1 | 1101.94 | 296 | 3.72 | 0.70 | 0.76 | 0.09 | 767.57 ** |
| 3 | Three-factor model 2 | 1126.90 | 296 | 3.81 | 0.69 | 0.75 | 0.09 | 792.53 ** |
| 4 | Three-factor model 3 | 896.72 | 296 | 3.03 | 0.76 | 0.82 | 0.08 | 562.35 ** |
| 5 | Two-factor model | 1643.68 | 298 | 5.52 | 0.55 | 0.60 | 0.11 | 1309.31 ** |
| 6 | One-factor model | 2056.63 | 299 | 6.88 | 0.44 | 0.48 | 0.13 | 1722.26 ** |

Notes: ** $p < 0.01$. Three-factor model 1: F1 + F2, F3, F4; three-factor model 2: F1, F2 + F3, F4; three-factor Model 3: F1, F2, F3 + F4; two-factor Model: F1 + F2, F3 + F4; one-factor model: F1 + F2 + F3 + F4.

The results of the correlational analyses indicate that creativity was positively related to both green recognition (r = 0.41, $p < 0.01$) and green disengagement (r = 0.21, $p < 0.01$). Green entrepreneurial intention was negatively related to green recognition (r = 0.37, $p < 0.01$) and positively related to green disengagement (r = −0.41, $p < 0.01$). Detailed results are shown in Table 2.

**Table 2.** Descriptive statistics and correlations.

| Variables | M | SD | 1 | 2 | 3 | 4 | 5 | 6 | 7 | 8 | 9 | 10 | 11 |
|---|---|---|---|---|---|---|---|---|---|---|---|---|---|
| 1. Age | 40.81 | 7.66 | | | | | | | | | | | |
| 2. Sex | 0.56 | 0.50 | −0.04 | | | | | | | | | | |
| 3. Edu | 0.75 | 0.43 | −0.03 | 0.07 | | | | | | | | | |
| 4. JT | 6.00 | 4.87 | 0.84 ** | −0.03 | −0.06 | | | | | | | | |
| 5. SE | 4.56 | 0.69 | 0.07 | −0.07 | 0.01 | 0.06 | (0.83) | | | | | | |
| 6. SC | 4.15 | 0.72 | 0.02 | −0.11 * | −0.05 | 0.03 | 0.36 ** | (0.83) | | | | | |
| 7. EC | 3.82 | 0.87 | 0.04 | −0.19** | 0.00 | 0.06 | 0.29 ** | 0.41 ** | (0.89) | | | | |
| 8. GSI | 4.83 | 0.94 | 0.14 ** | −0.13 * | −0.04 | 0.15 ** | 0.24 ** | 0.26 ** | 0.33 ** | (0.85) | | | |
| 9. GR | 4.63 | 0.93 | 0.08 | −0.11 * | −0.04 | 0.10 * | 0.22 ** | 0.19 ** | 0.41 ** | 0.17 ** | (0.87) | | |
| 10. GD | 3.49 | 0.95 | −0.07 | −0.05 | 0.02 | −0.07 | −0.09 | −0.05 | 0.21 ** | −0.07 | −0.18 ** | (0.85) | |
| 11. GEI | 4.70 | 1.05 | 0.05 | −0.09 | 0.00 | 0.06 | 0.26 ** | 0.28 ** | 0.21 ** | 0.38 ** | 0.37 ** | −0.41 ** | (0.84) |

Notes: N = 362. For gender, 0 = female, 1 = male. For education, 0 = high school degree or lower, 1 = university degree or higher. Age and job tenure units are presented in years. Reliability is shown on the diagonal. Edu = education; JT = job tenure; SE = self-efficacy; SC = social capital; EC = entrepreneurial creativity; GSI = green self-identity; GR = green recognition; GD = green disengagement; GEI = green entrepreneurial intention. * $p < 0.05$, ** $p < 0.01$.

### 4.2. Hypothesis Testing for the Mechanism of Green Recognition

To reduce multicollinearity, we initially made all the variables mean-centered. Table 3 shows the results of hierarchical multiple regression with green recognition as a mediator. In line with Hypothesis 1, Model 1 indicates that entrepreneurial creativity was positively associated with green recognition (B = 0.40, SE = 0.06, $p < 0.01$). In Model 3, we find that entrepreneurial creativity was not significantly associated with green entrepreneurial intention. This result further demonstrates that green entrepreneurial behavior is not solely a "business" issue. The result of Model 4 supports Hypothesis 2, showing that green recognition was positively associated with green entrepreneurial intention (B = 0.37, SE = 0.06, $p < 0.01$). Furthermore, Model 2 predicts that the interaction effect of entrepreneurial creativity and green self-identity is significantly associated with green recognition (B = 0.38, SE = 0.06, $p < 0.01$).

**Table 3.** Hierarchical regression analysis with green recognition as a mediator.

| Variables | DV = Green Recognition | | | | DV = Green Entrepreneurial Intention | | | |
|---|---|---|---|---|---|---|---|---|
| | Model 1 | | Model 2 | | Model 3 | | Model 4 | |
| | B | SE | B | SE | B | SE | B | SE |
| (Constant) | 0.00 | 0.04 | −0.10 * | 0.05 | 0.00 | 0.05 | 0.00 | 0.05 |
| Age | −0.00 | 0.01 | −0.00 | 0.01 | −0.00 | 0.01 | −0.00 | 0.01 |
| Gender | −0.04 | 0.09 | −0.02 | 0.09 | −0.09 | 0.11 | −0.08 | 0.10 |
| Education | −0.07 | 0.10 | −0.12 | 0.10 | 0.04 | 0.12 | 0.06 | 0.12 |
| Tenure | 0.02 | 0.02 | 0.02 | 0.02 | 0.01 | 0.02 | 0.00 | 0.02 |
| Self-efficacy | 0.15 * | 0.07 | 0.11 | 0.07 | 0.26 ** | 0.08 | 0.20 * | 0.08 |
| Social capital | −0.01 | 0.07 | −0.09 | 0.07 | 0.26 ** | 0.08 | 0.27 ** | 0.08 |
| EC | 0.40 ** | 0.06 | 0.29 ** | 0.06 | 0.09 | 0.07 | −0.06 | 0.07 |
| GSI | 0.01 | 0.05 | 0.12 * | 0.05 | | | | |
| EC × GSI | | | 0.38 ** | 0.06 | | | | |
| GR | | | | | | | 0.37 ** | 0.06 |
| R² | 0.19 | | 0.28 | | 0.12 | | 0.20 | |
| ΔR² | 0.17 | | 0.26 | | 0.10 | | 0.18 | |
| F value | 10.30 ** | | 14.95 ** | | 6.56 ** | | 10.97 ** | |

Notes. * $p < 0.05$, ** $p < 0.01$. EC = entrepreneurial creativity; GSI = green self-identity; GR = green recognition; GEI = green entrepreneurial intention.

Further, we conducted a simple slope test to examine the direction of the moderating effects on green self-identity. Figure 2 indicates that when green self-identity was high, higher creativity was significantly associated with higher green recognition (slope = 0.65, t = 9.57, $p < 0.01$), but when green self-identity was low, higher creativity was not significantly associated with lower green disengagement (slope = −0.06, t = −0.68, $p = 0.50$). Hence, Hypothesis 3 was partially supported.

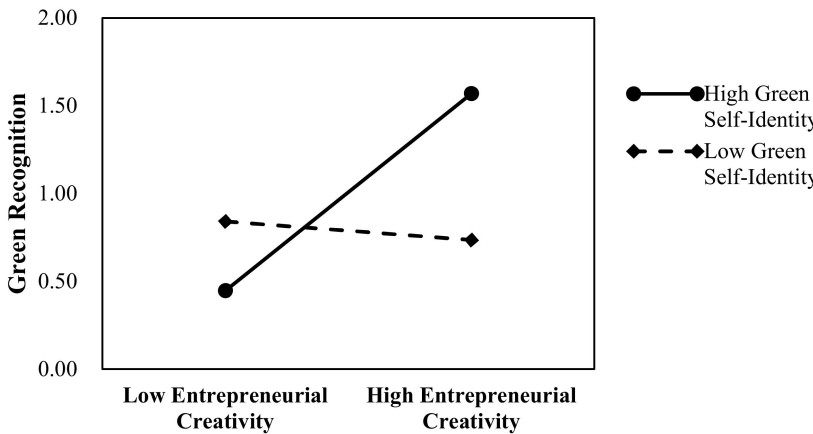

**Figure 2.** Interaction effect of entrepreneurial creativity and green self-identity on green recognition. *Notes.* The high and low levels of entrepreneurial creativity and green self-identity represent 1 *SD* above and below the mean, respectively.

We applied the PROCESS macro developed by Hayes [50] to test the conditional indirect effect of creativity on green entrepreneurial intention via green recognition. The results indicate that the conditional indirect effect of entrepreneurial creativity was significant for a high level of green self-identity (B = 0.24, SE = 0.04, $p < 0.01$) but not substantial for a low level of green self-identity. Hence, Hypothesis 4 was partially supported (see Table 4).

**Table 4.** Conditional indirect effect of entrepreneurial creativity on green entrepreneurial intention via green recognition.

| Green Self-Identity | The indirect Effect of Entrepreneurial Creativity × Green Self-Identity on Green Entrepreneurial Intention via Green Recognition | | | |
| --- | --- | --- | --- | --- |
| | B | SE | Boot LCI | Boot UCI |
| −1 SD | −0.02 | 0.04 | −0.11 | 0.03 |
| +1 SD | 0.24 | 0.04 | 0.15 | 0.32 |

*4.3. Hypothesis Testing for the Mechanism of Green Disengagement*

Table 5 shows the results of hierarchical multiple regression with green disengagement as a mediator. To avoid interference, we set age, gender, education, job tenure, self-efficacy, and social capital as the control variables. In line with Hypothesis 5, Model 1 indicates that entrepreneurial creativity was positively related to green disengagement (B = 0.36, SE = 0.06, $p < 0.01$). The results of Model 4 support Hypothesis 6, showing that green disengagement was negatively associated with green entrepreneurial intention (B = −0.49, SE = 0.05, $p < 0.01$). Furthermore, Model 2 predicts the interaction effect between entrepreneurial creativity and green self-identity to be significantly associated with green disengagement (B = −0.36, SE = 0.06, $p < 0.01$).

Further, we conducted a simple slope test to examine the directions of the moderating effects of green self-identity. Figure 3 indicates that when green self-identity was low, higher creativity was significantly associated with higher green disengagement (slope = 0.81, t = 7.40, $p < 0.01$), but when green self-identity was high, higher creativity was not significantly associated with lower green disengagement (slope = 0.12, t = 1.33, $p = 0.18$). Hence, Hypothesis 7 was partially supported.

**Table 5.** Hierarchical regression analysis with green disengagement as a mediator.

| Variables | DV = Green Disengagement | | | | DV = Green Entrepreneurial Intention | | | |
|---|---|---|---|---|---|---|---|---|
| | Model 1 | | Model 2 | | Model 3 | | Model 4 | |
| | B | SE | B | SE | B | SE | B | SE |
| (Constant) | 0.00 | 0.05 | 0.10 * | 0.05 | 0.00 | 0.05 | 0.00 | 0.05 |
| Age | −0.00 | 0.01 | −0.00 | 0.01 | −0.00 | 0.01 | −0.00 | 0.01 |
| Gender | −0.06 | 0.10 | −0.08 | 0.10 | −0.09 | 0.11 | −0.11 | 0.10 |
| Education | 0.02 | 0.11 | 0.07 | 0.11 | 0.04 | 0.12 | 0.05 | 0.11 |
| Tenure | −0.01 | 0.02 | −0.01 | 0.02 | 0.01 | 0.02 | 0.00 | 0.02 |
| Self-efficacy | −0.15 * | 0.08 | −0.12 | 0.07 | 0.26 ** | 0.08 | 0.17 * | 0.08 |
| Social capital | −0.16 * | 0.08 | −0.08 | 0.08 | 0.26 ** | 0.08 | 0.26 * | 0.06 |
| EC | 0.36 ** | 0.06 | 0.47 ** | 0.06 | 0.09 | 0.07 | 0.25 ** | 0.06 |
| GSI | −0.12 * | 0.06 | −0.22 ** | 0.06 | | | | |
| EC × GSI | | | −0.36 ** | 0.06 | | | | |
| GD | | | | | | | −0.49 ** | 0.05 |
| $R^2$ | 0.10 | | 0.18 | | 0.12 | | 0.29 | |
| $\Delta R^2$ | 0.08 | | 0.16 | | 0.10 | | 0.28 | |
| F value | 4.94 ** | | 8.46 ** | | 6.56 ** | | 18.28 ** | |

Notes. * $p < 0.05$, ** $p < 0.01$. EC = entrepreneurial creativity; GSI = green self-identity; GD = green disengagement; GEI = green entrepreneurial intention.

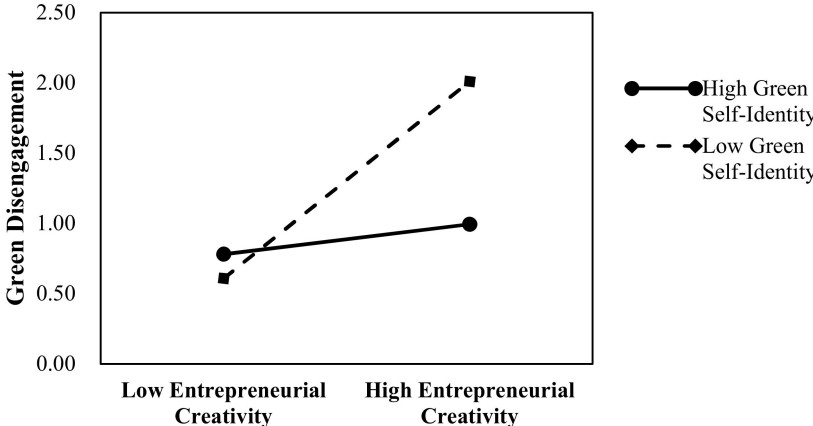

**Figure 3.** Interaction effect of entrepreneurial creativity and green self-identity on green disengagement. Notes. The high and low levels of entrepreneurial creativity and green self-identity represent 1 *SD* above and below the mean, respectively.

Furthermore, we applied the PROCESS macro by Hayes [50] to test the conditional indirect effect of entrepreneurial creativity on green entrepreneurial intention via green disengagement. The results indicate that the conditional indirect effect of entrepreneurial creativity was significant for a low level of green self-identity ($B = -0.39$, $SE = 0.08$, $p < 0.01$) but not substantial for a high level of green self-identity. Hence, Hypothesis 8 was partially supported (see Table 6).

**Table 6.** Conditional indirect effect of entrepreneurial creativity on green entrepreneurial intention via green disengagement.

| Green Self-Identity | The Indirect Effect of Entrepreneurial Creativity × Green Self-Identity on Green Entrepreneurial Intention via Green Disengagement | | | |
|---|---|---|---|---|
| | *B* | *SE* | Boot LCI | Boot UCI |
| −1 SD | −0.39 | 0.08 | −0.57 | −0.24 |
| +1 SD | −0.06 | 0.05 | −0.16 | 0.02 |

## 5. General Discussion

To promote the transition into a low-carbon economy, both practitioners and scholars are calling for more extensive research on green entrepreneurship [51,52]. By applying empirical analysis, the present study demonstrates that green recognition and green disengagement are two opposing mechanisms that either facilitates or inhibits green entrepreneurship, respectively. In particular, when the mechanism of green recognition is activated, creative entrepreneurs would be more likely to engage in green entrepreneurship; while when the mechanism of green disengagement is activated, creative entrepreneurs would be less likely to engage in green entrepreneurship. The result provides clues as to why creative entrepreneurs are sometimes not willing to start up a green venture (i.e., being "golden apples") and sometimes willing to start up a green venture (i.e., being "green apples"). Moreover, we also find that green self-identity works as a moderator that determines when creative entrepreneurs would be "golden apples" and when creative entrepreneurs would be "green apples". Specifically, creative entrepreneurs with high green self-identity are more likely to engage in green recognition and, thus, have more green entrepreneurial intention than those with low green self-identity. By contrast, creative entrepreneurs with low green self-identity are more likely to engage in green disengagement and, thus, have less green entrepreneurial intention than those with high green self-identity.

### 5.1. Theoretical Implications

The current research presents several theoretical implications by examining why and when creative entrepreneurs would be "golden apples" or "green apples"; in other words, why and when entrepreneurial creativity affects their green entrepreneurial intention. First, we used cognitive dissonance theory [10] to uncover the mechanisms of green recognition and green disengagement to influence green entrepreneurial intention. The existing literature proposes the key role of entrepreneurship on sustainable development of economics, society, and environment [53,54]. Our research extends the literature by introducing cognitive dissonance theory to propose a dual pathway model of green entrepreneurial intention. Specifically, to resolve the cognitive dissonance derived from green dilemmas, entrepreneurs may not only insist on green entrepreneurship through green recognition (Hypothesis 4) but also decline green entrepreneurship through green disengagement (Hypothesis 8). By proposing the dual pathway mechanism, we deepened our understanding of the cognitive processes related to green entrepreneurship.

Second, to examine the self-regulation perspective [9,30], we introduced a critical moderator, green self-identity, to examine when the two mechanisms are intensified or attenuated. We propose that green self-identity, an individual characteristic, is the underlying factor that determines the process of self-regulation [55]. The results show that creative entrepreneurs with high green self-identity are more likely to engage in green recognition than those with low green self-identity (Hypothesis 3). By contrast, creative entrepreneurs with low green self-identity are more likely to engage in green disengagement than those with high green self-identity (Hypothesis 7).

Third, our research extends creativity theory into the domain of green entrepreneurship. The existing literature has widely examined both the light [6,12,56,57] and dark side of creativity [41,42]. Our results show that entrepreneurial creativity is positively related, not only to the positive effect of green recognition, or the light side, but also to the negative effect of green disengagement, or the dark side. Moreover, although the present study does not provide evidence that entrepreneurial creativity is directly related to their green entrepreneurial intention, our results confirm that entrepreneurial creativity can result in either more or less green entrepreneurial intention under certain conditions.

### 5.2. Practical Implications

Our results provide evidence to help government agencies more accurately identify traditional commercial (i.e., golden apples) or green entrepreneurs (i.e., green apples). Specifically, creative entrepreneurs with high green self-identity are more likely to engage in green entrepreneurship, while

those with low green self-identity are more likely to engage in traditional commercial entrepreneurship. Furthermore, as an effect of self-regulation, government agencies could propose salient policies to help entrepreneurs identify the potential benefits of environmental businesses, such as financial support and favorable policies. In this way, entrepreneurs would have more chances to engage in green recognition and thus increase their green entrepreneurial intention. On the other hand, they could also use green-related policy advocacy to create a social atmosphere for environmental protection, such that a higher mental cost of green disengagement would be generated for entrepreneurs, thus favoring an increase in their green entrepreneurial intention.

### 5.3. Limitations and Future Research

While the present study proposes a dual pathway model of green entrepreneurial intention, it still has some limitations. For example, to minimize the influence of potential confounding factors, we did not consider any other personal characteristics (e.g., risk preference) that might result in diverse green decisions. It seems plausible that entrepreneurs with high-risk tendencies would be more likely to start green businesses, while those with a preference for low risk would be less likely to start green businesses. In future research, it would be worthwhile to explore personal characteristics other than green self-identity. Moreover, the self-reported measurement of the constructs in our work could also have resulted in some deviations from reflecting the true scenario. To increase the stability and robustness of our results, we suggest applying field and laboratory experiments in future studies.

**Author Contributions:** Conceptualization and writing—original draft preparation, H.J.; writing—review and editing, S.W.; methodology, L.W.; data curation, G.L. All authors have read and agreed to the published version of the manuscript.

**Funding:** This research was funded by National Natural Science Foundation of China with grant number 71704153, 71701180, 71704020 and China Postdoctoral Science Foundation with grant number 2018M642472.

**Conflicts of Interest:** The authors declare no conflicts of interest.

## Appendix A

**Table A1.** Items of the entrepreneurial creativity scale.

| Factor/Item |
| --- |
| 1. I can plan innovative entrepreneurial activities. |
| 2. I can plan entrepreneurial activities with my characteristics. |
| 3. I can plan stimulating entrepreneurial activities. |
| 4. Entrepreneurial activities that I plan are ingenious. |
| 5. Entrepreneurial activities that I plan are unique. |
| 6. Entrepreneurial activities that I plan are to guide the market. |
| 7. I understand customers' needs. |
| 8. I adapt practices flexibly to the changes. |
| 9. I consider preferences in the consumer market. |
| 10. Entrepreneurial activities that I plan are to meet customers' goals. |
| 11. Entrepreneurial activities that I plan can be adapted to different situations. |
| 12. Entrepreneurial activities that I plan are recognized in the consumer market. |

**Table A2.** Items of the green self-identity scale.

| Factor/Item |
| --- |
| 1. I have a strong sense of environmental management and protection. |
| 2. I have a sense of pride in my environmental goals and missions. |
| 3. I have carved out a significant position with respect to environmental management and protection. |
| 4. I have formulated a well-defined set of environmental goals and missions. |
| 5. I am knowledgeable about local environmental traditions and cultures. |
| 6. I identify strongly with others' actions with respect to environmental management and protection. |

**Table A3.** Items of the green recognition scale.

| Factor/Item |
| --- |
| 1. I can recognize new venture opportunities in environmental protection industries. |
| 2. I frequently identify ideas that can be converted into new products or services in environmental protection industries. |
| 3. I generally lack green ideas that may materialize into profitable enterprises. (reverse) |
| 4. I frequently identify opportunities to start up new businesses in environmental protection industries. |
| 5. I enjoy thinking about new ways of doing green businesses. |
| 6. I thought of many ideas for new green activities in the past month. |

**Table A4.** Items of the green disengagement scale.

| Factor/Item |
| --- |
| 1. It is okay for environmentalists to spread rumors to protect the environment. |
| 2. It is okay for environmentalists to gloss over specific facts to make the environmental point. |
| 3. Compared to other things people do that are not friendly to the environment, littering is not worth worrying about. |
| 4. You cannot blame people for environmental damage if that is what they were taught to do by their leaders. |
| 5. In contexts where nobody protects the environment, there is no reason to. |
| 6. It is okay for environmentalists to tell small lies when doing a work report of environmental protection because no one gets hurt. |
| 7. It is okay to treat badly somebody who behaves environmentally unfriendly. |
| 8. Non-environmentalists who get mistreated have usually done something to bring it on themselves. |

**Table A5.** Items of the green entrepreneurial intention scale.

| Factor/Item |
| --- |
| 1. I will do anything to become a green entrepreneur. |
| 2. My professional goal is to become a green entrepreneur. |
| 3. I will make every effort to establish and operate my own green business. |
| 4. I am seriously considering starting a green business. |
| 5. I am determined to become a professional green business manager. |
| 6. I am committed to developing my green business into a high-growth enterprise. |

**Table A6.** Items of the self-efficacy scale.

| Factor/Item |
| --- |
| 1. I can achieve most goals that I set for myself. |
| 2. When working on challenging tasks, I am sure that I will complete them. |
| 3. I can achieve outcomes that are important to me. |
| 4. I believe that I can succeed in most endeavors that I focus on. |
| 5. I can successfully overcome many challenges. |
| 6. I am confident that I can perform effectively in various tasks. |
| 7. Compared with other people, I can perform effectively in most tasks. |
| 8. I can perform effectively in difficult situations. |

**Table A7.** Items of the social capital scale.

| Factor/Item |
| --- |
| 1. There are several people I trust to help solve my problems. |
| 2. There is someone I can turn to for advice about making very important decisions. |
| 3. There is no one that I feel comfortable talking to about intimate personal problems. (reversed) |
| 4. When I feel lonely, there are several people I can talk to. |
| 5. If I needed an emergency loan of $500, I know someone I can turn to. |
| 6. The people I interact with online/offline would put their reputation on the line for me. |
| 7. The people I interact with would be good job references for me. |
| 8. The people I interact with would share their last dollar with me. |
| 9. I do not know people well enough to get them to do anything significant. (reversed) |
| 10. The people I interact with online/offline would help me fight an injustice. |

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
