# Peer review of "Golden Apples or Green Apples? The Effect of Entrepreneurial Creativity on Green Entrepreneurship: A Dual Pathway Model"

_sustainability, doi:10.3390/su12156285_

Round 1

Reviewer 1 Report

The paper “Golden Apples or Green Apples? The Effect of Entrepreneurial Creativity on Green Entrepreneurship: A Dual Pathway Model” is a well-written and interesting paper. It has the potential to add to our understanding on sustainable development in general and green entrepreneurship in particular. Hence, the paper has the potential to be published in Sustainability. Nevertheless, there is also room for improvement as more clarification is required of what is going on.

  1. Line 118-119 – authors can also refer to Patzelt and Shepherd, 2011. They have done a good job in explaining green/sustainable entrepreneurship. Also check Choongo el at., 2016 in sustainability.
  2. Line 267 – a new variable is introduced “social capital”. It is not clear why this variable is brought in the methods section when it was not discussed in the introduction and the theoretical background. It was not even hypothesized. More clarity is needed.
  3. The discussion is brief. Can it be enhanced? Also Golden Apples or Green Apples do not appear in the discussion. They are eye catching in the title and could be brought in the discussion and conclusion.

Reviewer 2 Report

This is a very interesting paper and it expands our current knowledge in the field by comparing creativity and green intention in entrepreneurship.

There two minor suggestions:

1) The introduction should not start with a story which is not stated the source and it does not offer something to readership. It is a little bit distractive.

2) The paper has to be enhanced with some more resources on green entrepreneurship and sustainable development. For example the following papers might be of use to enrich the discussion. 

Apostolopoulos, N., Al-Dajani, H., Holt, D., Jones, P., & Newbery, R. (2018). Entrepreneurship and the sustainable development goals. Emerald Publishing Limited.

Liargovas, P., Apostolopoulos, N., Pappas, I., & Kakouris, A. (2017). SMEs and green growth: The effectiveness of support mechanisms and initiatives matters. In Green economy in the Western Balkans. Emerald Publishing Limited.

Demirel, P., Li, Q. C., Rentocchini, F., & Tamvada, J. P. (2019). Born to be green: new insights into the economics and management of green entrepreneurship. Small Business Economics52(4), 759-771.

Mrkajic, B., Murtinu, S., & Scalera, V. G. (2019). Is green the new gold? Venture capital and green entrepreneurship. Small Business Economics52(4), 929-950
